# Reinforcement Learning for Contact-Rich Tasks: Robotic Peg Insertion Strategies

**Ang Zhang**
Department of Electronic Engineering
The Chinese University of Hong Kong
Shatin, Hong Kong
1155118934@link.cuhk.edu.hk

**Jianbang Liu**
Department of Electronic Engineering
The Chinese University of Hong Kong
Shatin, Hong Kong
1155071948@link.cuhk.edu.hk

## Abstract

Vision and touch are especially important when doing contact-rich manipulation tasks in unstructured environments. It is non-trivial to manually design a robot controller that combines these modalities which have very different characteristics. In this project,to connect vision and touch, we first equip robots with both visual and tactile sensors and collect a large-scale dataset of corresponding vision and tactile sequences. We use self-supervision to learn a compact and multimodal representation of our sensory inputs, which can then be used to improve the sample efficiency of our policy learning. We will train a policy in simulation environment using deep reinforcement learning algorithms.The learned policy is also transferable to handle real-world tasks. The peg insertion is chosen as the task for demonstration in this project. A preliminary version of our python implementation is available at: `https://github.com/Henry1iu/ierg5350_rl_course_project`. A video introducing our project is available at: `https://mycuhk-my.sharepoint.com/:v:/g/personal/1155071948_link_cuhk_edu_hk/EaKiGmkUvjJOoSqdWxrqjXYBpz3dCSAfOD9Co8krttyqUQ?e=RXsHD2`

## 1 Introduction

Humans perceive the world using multi-modal sensory inputs such as vision, audition, touch, taste, and smell. In routine tasks such as inserting a car key into the ignition, humans effortlessly combine the senses of vision and touch to complete the task. These two feedback modalities are complementary and concurrent during contact-rich manipulation. In this project, we plan to achieve a reinforcement learning policy that enables robot arm to perform some contact-rich manipulation tasks. As a case study, we choose the robotic peg insertion task, which is the most common problem of robotic assembly and the basis of a wide range of component assemblies (9). Our method starts with using neural networks to learn a good multi-modal representation of vision, touch and robotic proprioceptive information, including position, rotation, linear velocity and rotational angular velocity. Once we learn the representation, the resulting feature vector serves as input to a neural network policy trained through deep reinforcement learning. Figure 1 shows the construction of a robotic peg-in-hole system briefly.

## 2 Related Work

Contact-rich tasks, such as peg insertion, fastening screws, and edge following, have been studied for decades due to their relevance in manufacturing (3). Manipulation policies often rely entirely on haptic feedback and force control, and assume sufficiently accurate state estimation (9). Some combine visual and haptic data for inserting two planar pegs with more complex cross sections, but assume known peg geometry (7).

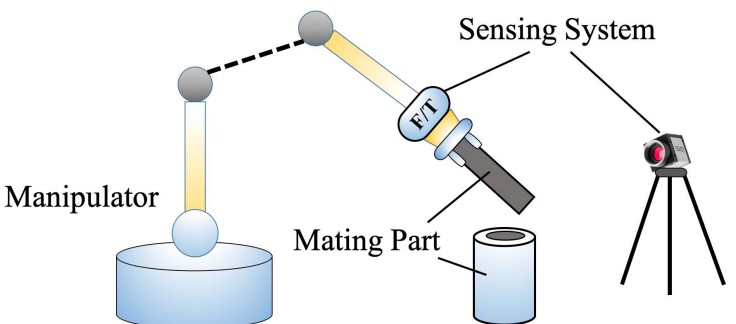

Figure 1: Setup of the robotic peg-in-hole system

Recently, reinforcement learning has been proposed to handle variations in geometry and configuration for manipulation. However, fewer approaches exploit the complementary nature of vision and touch. Some of them perform full manipulation tasks based on multiple input modalities, but require a pre-specified manipulation graph (1). For reinforcement learning, a good representation encodes the essential information of the state for the agent to choose its next action for a given task. A compact and low-dimensional state representation can make reinforcement learning more data efficient. Similar to (4), we also adopt a self-supervised objective to fuse visual and haptic data by predicting whether visual and haptic data are temporally aligned.

## 3 Simulation Environment and Problem Formulation

The goal of this project is to design a framework to train a policy in simulation environment for a robot arm to conduct some contact-rich manipulation tasks. In addition, The learned policy is also transferable to handle real-world tasks. Specifically, the peg insertion is chosen as the task for demonstration. The action space of the robot arm is constrained to a continuous space with 3-DoF, where the robot arm can only operate in Cartesian mode(move the end-effector position in Cartesian coordinate) and the orientation of the end-effector remains the same during the operation.

We construct a simulation environment using PyBullet library. In the environment, a Kuka LBR iiwa robot arm is installed on a table along with the target box (see Figure 2). The target box (see Figure 3a) and the end-effector (see Figure 3b) is modeled in SolidWorks and imported into the environment by configuring a URDF file. At each step, the environment will generate a $224 \times 224$ RGB color image (see Figure 4a), a $224 \times 224$ depth image(see Figure 4b) and the force torque (F/T) reading (see Figure 5) captured at the joint connected with the end-effector. The F/T reading is composed of six values indicating the value of force or torque sensed by the joint along each axis in its Cartesian coordinate.

As in (3), the contact-rich tasks are modeled as a finite-horizon, discounted Markov decision process $\mathcal{M}$. Given the action space $\mathcal{A}$, the state space $\mathcal{S}$, the state transition dynamics $\mathcal{T} : \mathcal{S} \times \mathcal{A} \to \mathcal{S}$, the optimal stochastic policy $\pi : \mathcal{S} \to \mathbb{P}(\mathcal{A})$ can be determined by maximizing the expected discounted reward $R(\pi)$:

$$R(\pi) = \mathbb{E}_\pi[\sum_{t=0}^{T-1} \gamma_d^t r(\mathbf{s}_t, \mathbf{a}_t)] \tag{1}$$

where $\mathbf{s}_t$ represents the state at time step $t$, $\mathbf{a}_t$ represent the action taken at $\mathbf{s}_t$, $\gamma_d \in (0, 1]$ is the discount factor, and $T$ is the horizon.

We define our policy with a neural network $\theta_\pi$, which contains a state encoder and a three-layer Multi-layer Perception(MLP) network. The encoder will take the multi-modal input and predict the state vector. The MLP will take in the state and generate a 3D displacement of the end-effector.

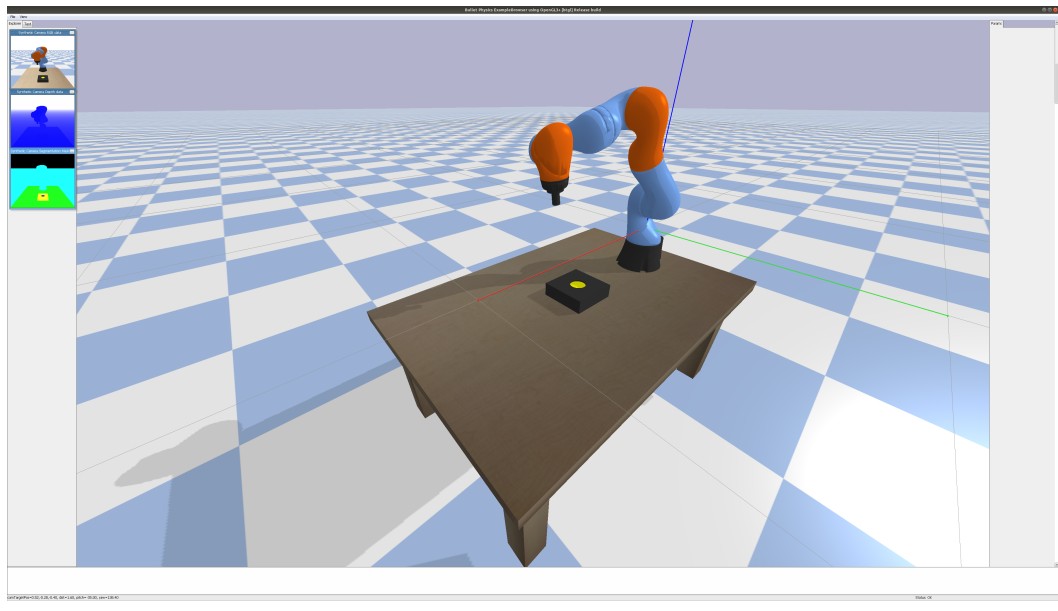

Figure 2: The simulation environment: The robot arm and the insertion target.

Currently, we implement the reward function defined as following:

$$r(\mathbf{s}) = \begin{cases} 10 & \text{if box is reached} \\ 50 & \text{if insertion completion} \\ -50 & \text{if table is reached} \\ -\|\mathbf{s}_e - \mathbf{s}_b\| & \text{if neither reaching nor completion} \end{cases} \tag{2}$$

where $\mathbf{s}_e = (x_e, y_e, z_e)$ represents the end-effector's position. The goal position of the end-effector is $\mathbf{s}_b = (x_b, 0, z_b)$, which is the center of the bottom inside the box.

## 4  Framework

In order to solve the problem using reinforcement learning, we implemented a on-policy learning algorithm named proximal policy optimization (PPO), which is a improved version of trust region policy optimization (TRPO). We implement a Actor-Critic policy model for the agent. The incoming multi-modal observation will be analysed by a multi-modal fusion encoder to generate a compact state

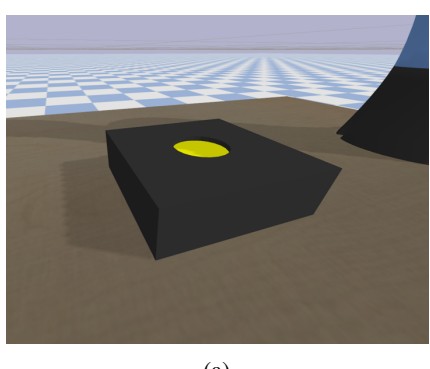
(a)

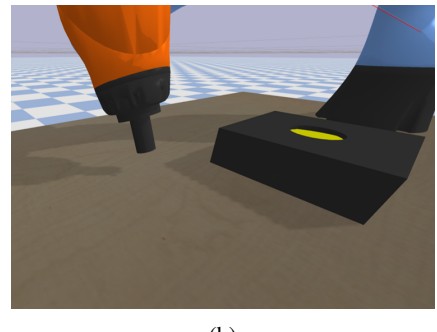
(b)

Figure 3: The (a) 3D printed target box with the target hole and (b) a close look of peg and the target box.

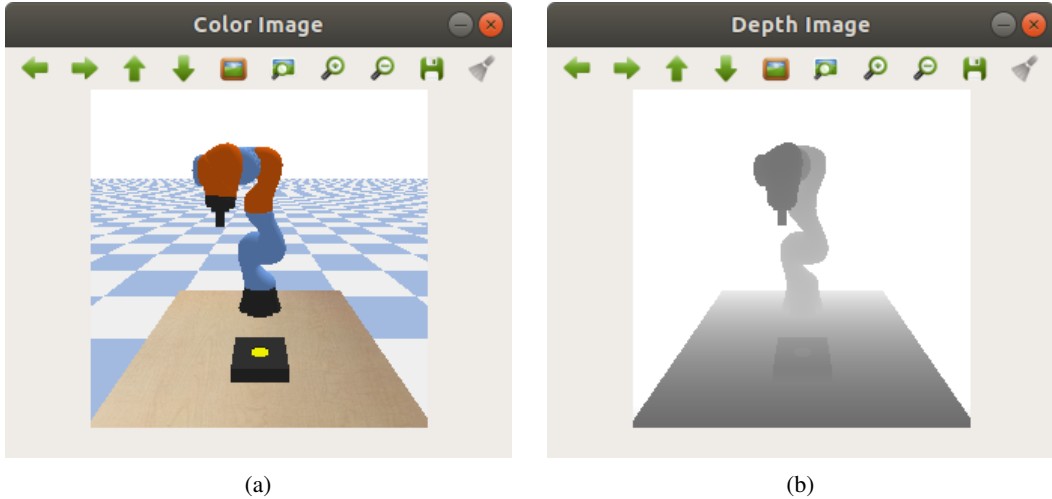

Figure 4: The example of (a) the color image and (b) the depth image.

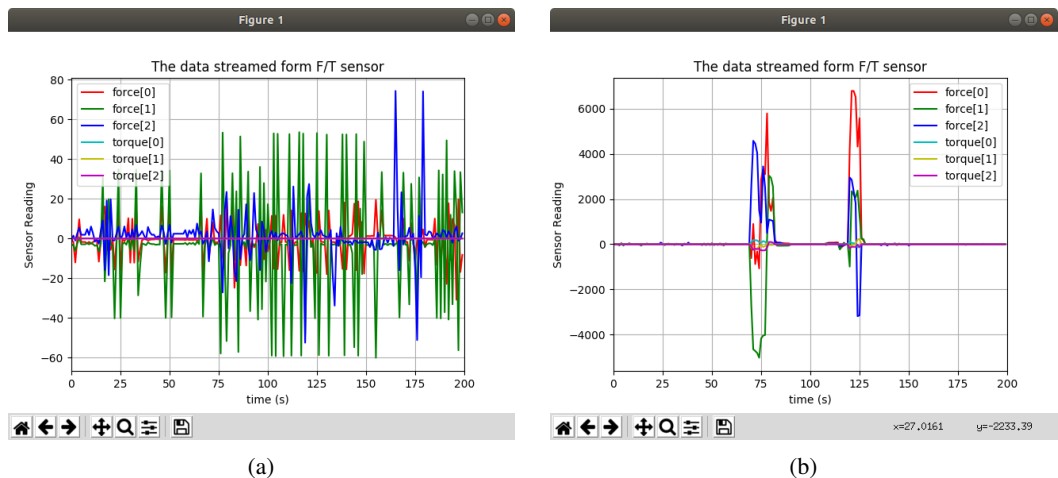

Figure 5: The visualization of F/T readings in a sliding window form (a) while the robot arm is moving without touching anything; (b) when the end-effector hits or touches a object during its motion.

representation. Then, the state representation will be taken by the actor and critic model for making the action decision and predicting the state value corresponding to current policy. However, with our design, the policy model become deeper and more complicated, leading to a more computationally expensive optimization and higher chance of over-fitting. Inspired by (3), we designed a multi-branch auto-encoder structure for learning the fused representation of visual and haptic sensor data from massive data samples. After training with realistic or simulated visual and haptic data, the modality encoders can concentrate on the key features and generate a compact representation of the MDP state. Collaborating a well pre-trained encoders, the training of the policy network $\theta_\pi$ is expected to converge faster with resulting a better mean reward.

## 4.1 Reinforcement Learning

Our final goal is to equip a robot with a policy for performing contact-rich manipulation tasks. It is desirable to enable a robot to learn control policies through trial-and-error, where the learning process is applicable to a broad range of tasks. In this project, we use a peg insertion task as our evaluation task. Given its recent success in continuous control (2)(6), deep reinforcement learning lends itself well to learning policies that map high-dimensional features to control commands.

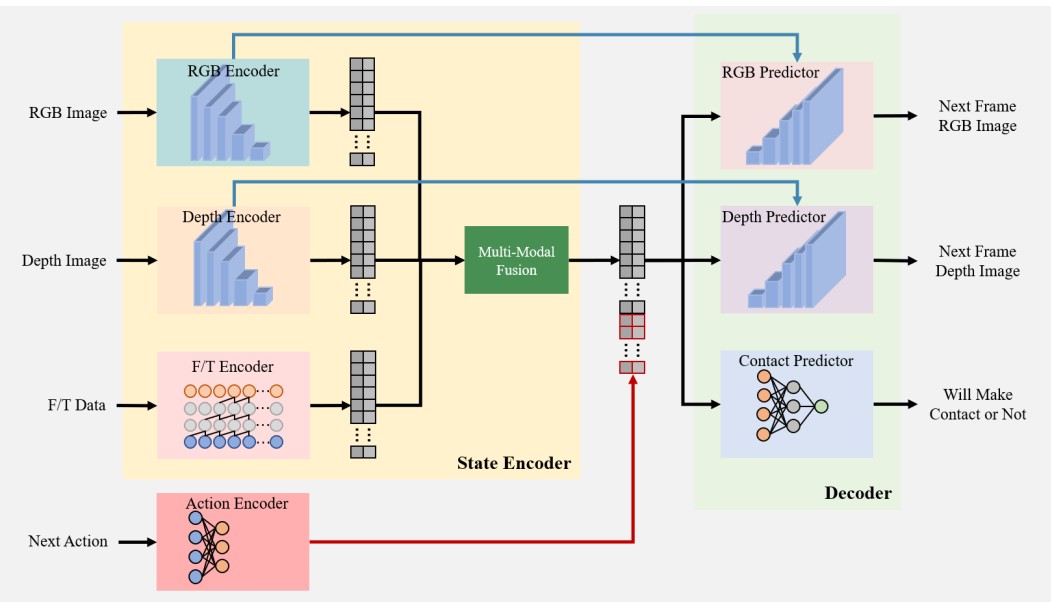

Figure 6: The illustration of the model structure designed to learn the multi-modal representation.

Modeling contact interactions and multi-contact planning still result in complex optimization problems that remain sensitive to inaccurate actuation and state estimation. We formulate our peg insertion task as a model-free reinforcement learning problem to investigate its performance when relying on multimodal feedback. By choosing model-free, we also eliminate the need for an accurate dynamics model, which is typically difficult to obtain in the presence of rich contacts. Specifically, we plan to choose proximal policy optimization algorithm (PPO), which is a policy gradient method (6). PPO proposes clipped surrogate objective and adaptive KL penalty coefficient to solve constrained optimization. We plan to design a policy network that takes the d-dimensional multimodal representation as input and produces 3D position displacement $\Delta x$ of the robot end-effector.

## 4.2 Multi-modal Sensing and MDP Status Representation

The structure of the multi-modal is shown in Figure. 6. There are 3 input branch and 3 different encoders handling different types of sensory data. The encoded sensory data representation will be fused into a single multi-modal representation in the fusion module. The 3 decoder branches designed based on various objectives will predict corresponding results.

**Encoding**: Firstly, the various types of data will be encoded individually with different encoders. The encoders for the visual data (color image and depth image) are similar. The higher dimensional features of the RGB and depth images will be extracted by a backbone consisting of 6 convolution layers. Later, two fully-connected layer will transform the features into a $2 \times 128$-dimensional variational parameter vector. For the haptic data, the last 32 reading from the sending the take-action command is taken. This $32 \times 6$ torque data is performed 5-layer causal convolution (5) and also transformed into a $2 \times 32$-dimensional variational parameter vector.

**Multi-Modal Fusion**: According to (8; 4), we assume every encoder maps to a multivariate isotropic Gaussian prior and each modality is conditionally independent. The encoders' outputs is taken as the mean and variance, each has a dimension of $1 \times 128$. The multivariate Gaussian distribution of the multi-modal latent space can be computed by combining the modality-specific distribution as follows:

$$\sigma_j^2 = \left(\sum_{i=1}^{n+1} \sigma_{ij}\right)^{-1} \qquad \mu_j = \left(\sum_{i=1}^{n+1} \mu_{ij}/\sigma_{ij}^2\right)\left(\sum_{i=1}^{n+1} \sigma_{ij}^2\right)^{-1} \tag{3}$$

where $n$ is the number of modalities, $\mu_j$ and $\sigma_j^2$ represents the variational parameters of the $j$th dimension of the posterior distribution encoded by the encoder. During the state representation

learning, the multi-modal fusion module produce the variational parameters. During the inference, the module will produce the $1 \times 128$ state representation, respectively.

**Self-Supervised Decoding**: Similarly, we carefully design the learning objectives to ensure all the ground truth for the prediction can be determined automatically. The idea of self-supervise learning saves tremendous efforts in annotating the data for training. The decoders will predict the color and depth image frame after the action is executed and whether the peg makes contact with the target based on the taken action. As illustrated in Figure 6, the action taken by the robot arm will be encoded by the action encoder, which is a 2-layer MLP. The encoded action feature along with the multi-modal representation will be processed by the 3 decoder branches to predict corresponding results. Since the state representation is over simplified for the image prediction, the multi-scale feature maps extracted by different encoder layers will be passed to the encoder to facilitate the prediction, as illustrated by the blue shortcut path. The image frame predictors will generate a predictive images by merging the multi-scale feature maps and gradually recovering the spatial resolution. The contact predictor will make a guess on whether the end-effector will touch the target box.

## 5    Current Progress

### 5.1    Multi-Modal State Representation Pre-train

While collecting the multi-modal data for representation learning, We deploy a random policy. The robot arm move randomly in the action space constrained within a cube whose vertical center and bottom is aligned with the target box. Then, the self-annotated ground truth can be obtained using some straightforward criteria. The contact ground truth can be easily determined by monitoring the reward, a positive reward means the target box is touched. Aiming at training a stable state representation encoder, we collected 25000 samples collected with the random sampling, 20000 for training and 5000 for evaluation. Each sample contains the color image, depth image and F/T reading before a action is taken as well as the taken action, color image, depth image and the contact label.

We train the end-to-end network structure on a desktop with Intel Core(TM) i7-10700K@3.80GHz processors, 64G DDR4 Memory and a NVIDIA GeForce RTX 2080 (8 GB) installed. In each training process, the network is trained for 50 epochs, where the model with the smallest evaluation loss is selected. The Adam optimizer is adopted with a initial learning rate of 3e-3 and a weight decay of 5e-4.

### 5.2    Reinforcement Learning

After training with simulated visual and haptic data, we obtain a compact multimodal representation from the modality encoders. Then we implement the PPO method using PyTorch on the Kuka peg-in-hole environment. This model is trained on a Nvidia 2080ti GPU with a initial learning rate of 1e-3, one parallel environment and a entropy loss weight of 0.01. The number of sampled steps is one million and the mean reward in final is about between -150 to -200.

The model that doesn't use the pre-trained encoder is also trained as a comparison test. In this model, the network will learn the parameters of both encoders and Actor-Critic policy model. It needs real-time sampling and online learning, which will lead a more computationally expensive optimization and higher chance of over-fitting. As shown in figure 7, the model without pre-trained encoder quickly falls into a local minimum, and the reward stabilizes at -340. Compared with it, the model we proposed using a pre-trained encoder has a better performance. We can find the reward can reach -110 at most. It is a negative reward because we set the penalty higher (-50 if table is reached). Using this model, the kuka robot can reach the box after some attempts.

## 6    Conclusion

In this project, we explored the effectiveness of joint reasoning over time-aligned multimodal results in contact-rich tasks. We built our simulation environment for collecting data and the online training. We designed a multi-branch auto-encoder structure for learning the fused representation of visual and haptic sensor data from massive data samples. We also applied the self-supervision to the model that trains the representation, which can eliminate the need for manual annotation. After pre-trained, the

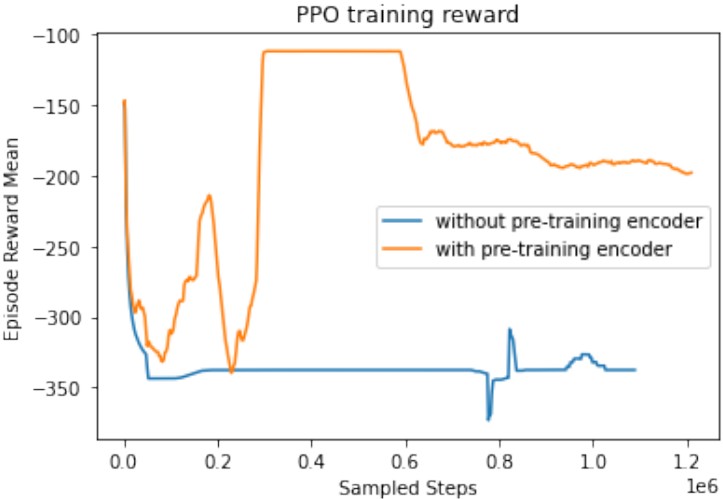

Figure 7: PPO training reward in kuka Peg Insertion

multimodal representation encoder will be deployed in the PPO policy for reinforcement learning. Through the contrast test, we can find the pre-trained encoder can improve the performance of reinforcement learning. Although the robot didn't finish the insertion task, it can touch the box after some attempts. It is valuable and effective to learn manipulation policy through reinforcement learning combining the multi-sensory information.

# 7   Future Work

Although, the multi-modal state representation pre-train help stabilize the online reinforcement learning. The trained policy didn't acquire the ability of inserting the peg into the hole accurately. We are considering several enhancement:

1. Try an off-policy learning algorithm, such as Twin-Delayed DDPG.

2. Revise the reward function: By returning a punishment (negative reward) correlated with the distance between the position of the end-effector and the target, it is still too abstract for the agent to learn how to approach the target through a shortest feasible trajectory. Turning those discredited reward (reaching and completion reward) into a continuous reward function may help the agent better understand the goal.

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
