# OpenReview forum: "Reinforcement Learning for Contact-Rich Tasks: Robotic Peg Insertion Strategies"
_CUHK.edu.hk/2021/Course/IERG5350_

### Official Review · AnonReviewer1 · 2020-12-15
**Valuable and Significance paper, but could be better if needed to reach the top**

**Rating:** 8
**Confidence:** 3

**Review:**

I am not very familiar with the relevant literature, also I am not the profession in Mechanical Engineering (as this is related to this), but I tried to make my review honestly based on the report and what I know.

Significance: This is the paper to investigate Contact-Rich Tasks with the application of RL. I think this is significance to the industrial development, as this can reduce the production cost, manpower and time required in some precision industrial process in the near future. Just like Robotic Peg Insertion that investigated in this paper, this can apply to drilling to find the best place to drill in order to enhance mechanical performance of the machine operation.

Novelty: This is obvious an innovation and may be able to apply in the Industrial 4.0

Technical quality: Good. Clear defined formula and methodology.

Clarity: It is clear to read, well-organized and able to follow your logic. Especially the figures that provided by you guys.

My some comments or suggestions:

(1) Is it a typo? (p.6, RTX2080Ti or RTX2080 to train the model? Or you will 2 GPUs to do the 2 parts of training, based on your logic?)

(2) It will be good to have more analysis in the current result. (e.g. Why the PPO training reward in kuka Peg Insertion is like this?)

(3) Of course your future work, using off-policy alg may have the positive effect to your agent, and I hope I can have chance to see such things if you have time as it is valuable.

---

### Official Review · AnonReviewer3 · 2020-12-18
**This paper is innovative, significant and easy to read. It would be better if there are more discussion about the results.**

**Rating:** 8
**Confidence:** 4

**Review:**

Probs:

1.Significant: This topic is significant and worth exploring. With the advantages of low self weight, high flexibility, easy programming, rapid configuration and small operation space limitations, robotic peg also has broad application prospects in medical, education and training, new retail and other service industry scenarios.

2.Clarity: This paper has good structure and grammar, which is easy to read. The reader can get the idea of the author easily.

3.Originality: There are many implement details in the paper, which convinces me that the work is done by authors rather than others.

4.Quality: The authors apply what they have learned in this course (PPO) to automatic controlling problem. This is creative and Innovative, especially they uses autoencoder to extract the vision. Their work also improves the quality of PPO.

Cons:
1. It didn't compare the model with contact-rich tasks that don't use reinforcement learning and the robot cannot achieve the anticipated goal---insertion. But this is qualified for course project.
2. It didn't  analyze the final result.

Question and suggestion:
1. How to make sure the encoder extract the best representation?

---

### Official Review · AnonReviewer2 · 2020-12-19
**Nice work but could be better**

**Rating:** 7
**Confidence:** 4

**Review:**

General:

Significance: This paper uses proximal policy optimization (PPO) to learn robotic peg insertion strategies, uses PyBullet library to construct a simulation environment, and compares the experimental results between the proposed pre-train model and the proposed model without pre-train.

Novelty: Using reinforcement learning to learn robotic strategies is not new, but using PPO to learn robotic peg insertion strategies is innovative.

Technical quality:  This paper use PPO to solve the focused problem and conduct the experiments in a simulation environment.

Clarity: The clarity is good and easy to follow. This paper first defines the robotic peg insertion problem, then defines the expected discounted reward function and the reward function. Suitable pictures help me understand this paper better.

Specific:

Pros: a. define the robotic peg insertion problem clearly; b. use PPO to solve the problem; c. define the corresponding reward function.

Cons: a. you can do more experiments to further investigate the proposed method since there are only one picture and no table that show the experiment results; b. the discussion about the experiment results should be longer; c. as you said in the future work part, try an off-policy learning algorithm, and maybe you can check this paper: https://arxiv.org/pdf/2008.10224.pdf , which uses SAC to solve the robotic peg-in-hole problem.